sv-callers: a highly portable parallel workflow for structural variant detection in whole-genome sequence data

http://orcid.org/0000-0003-1711-7961 Kuzniar Arnold 1 a.kuzniar@esciencecenter.nl
http://orcid.org/0000-0002-8172-4865 Maassen Jason 1
http://orcid.org/0000-0002-5821-2060 Verhoeven Stefan 1
http://orcid.org/0000-0001-8784-2507 Santuari Luca 2
Shneider Carl 2
http://orcid.org/0000-0003-3357-4580 Kloosterman Wigard P. 2
http://orcid.org/0000-0002-0828-3477 de Ridder Jeroen 2 j.deridder-4@umcutrecht.nl
1 Netherlands eScience Center , Amsterdam , Netherlands
2 Center for Molecular Medicine, University Medical Center Utrecht , Utrecht , Netherlands
Gillespie Joseph
Electronic publication date: 2020 Jan 6
Publication date: 2020
Volume: 8
Electronic Location ID: e8214
Received 2019 May 15; Accepted 2019 Nov 14
Copyright: © 2020 Kuzniar et al.
Copyright year: 2020
Copyright holder: Kuzniar et al.
License: This is an open access article distributed under the terms of the Creative Commons Attribution License, which permits unrestricted use, distribution, reproduction and adaptation in any medium and for any purpose provided that it is properly attributed. For attribution, the original author(s), title, publication source (PeerJ) and either DOI or URL of the article must be cited.
License URL: https://creativecommons.org/licenses/by/4.0/

Keywords: Scientific workflow, Cancer genomics, Variant calling, Structural variants, High-performance computing, Cloud computing, Snakemake, Xenon, Research software, Open science

Funding: The Netherlands eScience Center 027016G03 Dutch National e-infrastructure with the support of SURF Foundation (NWO) 16669 This work was supported by the Netherlands eScience Center (Grant Number: 027016G03) and was carried out on the Dutch national e-infrastructure with the support of SURF Foundation (NWO Project Number: 16669). The funders had no role in study design, data collection and analysis, decision to publish, or preparation of the manuscript.

==============================
Structural variants (SVs) are an important class of genetic variation implicated in a wide array of genetic diseases including cancer. Despite the advances in whole genome sequencing, comprehensive and accurate detection of SVs in short-read data still poses some practical and computational challenges. We present sv-callers, a highly portable workflow that enables parallel execution of multiple SV detection tools, as well as provide users with example analyses of detected SV callsets in a Jupyter Notebook. This workflow supports easy deployment of software dependencies, configuration and addition of new analysis tools. Moreover, porting it to different computing systems requires minimal effort. Finally, we demonstrate the utility of the workflow by performing both somatic and germline SV analyses on different high-performance computing systems.

Introduction

Structural variants (SVs) such as deletions, insertions and duplications account for a large part of the genomic diversity among individuals and have been implicated in many diseases including cancer. With the advent of novel DNA sequencing technologies, whole genome sequencing (WGS) is becoming an integral part of cancer diagnostics that can potentially enable tailored treatments of individual patients (Stratton, 2011). Despite advances in large-scale cancer genomics projects (such as the TCGA and PCAWG of the International Cancer Genome Consortium; https://icgc.org/), systematic and comprehensive analysis of massive genomic data, in particular the detection of SVs in genomes, remains challenging due to computational and algorithmic limitations (Alkan, Coe & Eichler, 2011; Yung et al., 2017; Ma et al., 2018; Gröbner et al., 2018).

Recent tools for somatic and germline SV detection (callers) exploit more than one type of information present in WGS data (Lin et al., 2015). For example, DELLY (Rausch et al., 2012) relies on split reads and discordant read pairs while LUMPY (Layer et al., 2014) additionally utilizes read depth information. Furthermore, callers such as Manta (Chen et al., 2016) and GRIDSS (Cameron et al., 2017) also incorporate short-read assembly. To obtain a more comprehensive and/or accurate callset, ensemble approaches have yielded promising results (English et al., 2015; Mohiyuddin et al., 2015; Becker et al., 2018; Fang et al., 2018). In such an approach, (i) a range of SV callers are executed, and (ii) their results are combined into a single callset. While this approach has been demonstrated to improve SV callsets, the step (i) poses a major bottleneck as running multiple SV callers efficiently on the user’s computational infrastructure and/or adding new SV callers (as they become available) is far from straightforward.

A common practice to couple multiple tools together is by monolithic “wrapper” scripts (English et al., 2015; Fang et al., 2018) and, to a lesser extent, by a workflow system. The latter is a recommended approach to improve the extensibility, portability and reproducibility of data-intensive analyses (Leipzig, 2017). Specifically, multiple command-line tools, usually written in different languages, are integrated in a workflow (instance) that is defined either by an implicit syntax (Köster & Rahmann, 2012; Holmes & Mungall, 2017) or by an explicit syntax (Afgan et al., 2018; Vivian et al., 2017). In addition, a workflow system might support parallel execution of tasks through a batch scheduler that distributes the tasks across nodes of a high-performance computing (HPC) system. For example, Snakemake is an established generic workflow system inspired by the GNU Make build automation tool (Köster & Rahmann, 2012). It requires a “recipe” with implicit rules to generate output from input files. Moreover, Snakemake supports Conda package manager and light-weight containers such as Docker (https://www.docker.com/) or Singularity (Kurtzer, Sochat & Bauer, 2017), Common Workflow Language (Amstutz et al., 2016) and parallel execution in HPC cluster (via DRMAA, http://www.drmaa.org/) or cloud environments (via Kubernetes, https://kubernetes.io/). Furthermore, a workflow developed on one system is not necessarily portable to or easily reusable on another system due to the complexity of software environments, system usage policies (e.g., lack of root privilege or absence of Docker) and/or the use of different batch schedulers. Typically, compute clusters rely on batch schedulers such as Grid Engine, Slurm or Torque to manage resources and to distribute tasks over the available nodes. As a result, workflows need to be adapted to use scheduler- or file transfer-specific commands, which makes the distribution of compute tasks across (heterogeneous) systems cumbersome.

To alleviate these problems, we developed sv-callers, a user-friendly, portable and scalable workflow based on the Snakemake and Xenon (middleware) software. The workflow includes state-of-the-art somatic and germline SV callers, which can be easily extended, and runs on HPC clusters or clouds with minimal effort. It supports all the major SV types (i.e., deletions, insertions, inversions, duplications and translocations) as detected by the individual callers. In this paper, we focus on the implementation and computational performance of the workflow as well as provide examples to improve the reproducibility of downstream analyses.

Methods

Experimental data and setup

The sv-callers workflow was tested with two human WGS datasets, one for the single-sample mode (i.e., germline SV detection) and the other for the paired-sample mode (i.e., somatic SV detection in tumor with matched normal). The benchmark data were obtained by Illumina HiSeq 2500 sequencing of the NA12878 genome (BioProject:PRJNA200694, ENA:SRX1049768–SRX1049855, 148 GiB BAM file downsampled to 30× coverage) released by the NIST Genome in a Bottle Consortium (Zook et al., 2016). The cell lines data were obtained by Illumina HiSeq X Ten sequencing of a cancer cell line derived from malignant melanoma (COLO829, EFO:0002140) at 90× coverage (ENA:ERX2765496, 184 GiB BAM file) and a matching control, lymphoblastoid cell line (EFO:0005292) at 30× coverage (ENA:ERX2765495, 67 GiB BAM file). Illumina short reads were mapped to the human reference genome (i.e., b37 and GRCh37 were used for the benchmark and the cell lines data, respectively) using the BWA-MEM algorithm (Li, 2013). To increase the amount of jobs, the WGS samples in BAM files were copied ten times, resulting in 140 jobs per dataset (of which 80 and 60 were SV calling and post-processing jobs, respectively). This allowed us to reliably estimate compute resources used as well as to assess if the results are correct and/or reproducible across the sample copies using the UpSet(R) plots (Conway, Lex & Gehlenborg, 2017).

Structural variation detection

Four SV callers were included in the workflow namely Manta (v1.1.0), DELLY (v0.7.7), LUMPY (v0.2.13) and GRIDSS (v1.3.4), taking into account the state-of-the-art methods (e.g., used in the ICGC projects), active development and software/documentation quality. Moreover, each of the callers have been extensively validated in their respective publications. Most callers make use of more than one programming language and take advantage of multi-core architectures (Table 1). Further, we accommodated the Conda package manager (https://conda.io/) including the ‘bioconda’ channel (Da Veiga Leprevost et al., 2017) to ease the installation of required bioinformatics software. Structural variant detection involved the callers’ default settings and ‘best practices’ for post-processing the callsets in VCF/BCF files: (i) genomic regions with anomalous mappability (ENCODE:ENCFF001TDO in BED file) were filtered using the SURVIVOR software (v1.0.6; Jeffares et al., 2017) with filter sub-command, (ii) low-quality calls and/or (iii) germline calls in the paired-sample (somatic) mode were filtered using BCFtools (v1.9; Li et al., 2009), and finally (iv) the resulting SV calls of the four detection tools were merged into one (union) set using SURVIVOR with merge sub-command with user parameters defined in the workflow configuration file (analysis.yaml). In our example analyses (Kuzniar & Santuari, 2019), two SVs were considered overlapping and merged into one entry if the distance between the relative breakpoints at each end is smaller than 100 bp, regardless of the SV type. Note that this threshold is within the estimated insert sizes of the benchmark (552 bp) and the cell lines (518/531 bp for tumor/normal) datasets. Furthermore, the germline SV calls of each tool including the derived (merged) callsets were evaluated in terms of precision (TP/(TP + FP)) and recall (TP/(TP + FN)) using the StructuralVariantAnnotation R package (v1.0.0; Cameron & Dong, 2019) with high-confidence deletions (truth sets) published by Layer et al. (2014) (PacBio/Moleculo data, n = 4,095) and Parikh et al. (2016) (Personalis/1000 Genome Project data, n = 2,676).

Table 1 SV detection tools included in the workflow.

Software	Implementation	Parallelism	I/O files	
Manta	C++, Python	pyFlow tasks1, SIMD2	FASTA, BAM or CRAM/VCF	
DELLY	C++	OpenMP threads3	FASTA, BAM/BCF	
LUMPY	C/C++, Python	–	FASTA, BAM/VCF	
GRIDSS	Java, R, Python	Java threads, SIMD2	FASTA, BAM or CRAM/VCF or BCF	
Notes:

1 http://illumina.github.io/pyflow.

2 Single instruction multiple data vector processing.

3 Depends on the number of input samples used (i.e., max. two in the paired-sample mode).

Workflow implementation and deployment

We implemented the sv-callers workflow (v1.1.0; Kuzniar et al., 2018; Kuzniar, 2019) using the Snakemake workflow system (v4.8.0; Köster & Rahmann, 2012) and the Xenon software suite (v3.0.0). Figure 1A shows a schematic diagram of the workflow including input/output files. To perform an analysis, a user provides a CSV file with the (paired) samples and adjusts the workflow configuration file(s) accordingly. The file analysis.yaml is concerned with the analysis itself (such as the mode, reference genome, exclusion list, callers, post-processing, resource requirements, etc.) while the file environment.yaml includes software dependencies/versions. Moreover, the workflow parameters are also accessible through the command-line interface (see snakemake -C [params] argument).

Figure 1 Workflow/Xenon software architecture.

(A) The workflow includes four SV callers namely Manta, DELLY, LUMPY and GRIDSS to detect SVs in (paired) samples given a reference genome. Submitted jobs execute in parallel across compute nodes. SVs detected per caller are filtered and merged into one (union) call set. Note: DELLY requires separate runs for each SV type (i.e., DUP, duplication; DEL, deletion; INS, insertion; INV, inversion; BND, translocation). (B) The workflow uses the Xenon command-line interface (xenon-cli) that abstracts away scheduler- and file transfer-specific commands; the Xenon library translates each command to the one used by the target system.

We developed the Xenon library (Maassen et al., 2018) and the associated command-line interface, xenon-cli (Verhoeven & Spaaks, 2019) to provide uniform access to different batch schedulers and file transfer clients. Xenon is comparable to the Simple API for Grid Application (SAGA) (Merzky, Weidner & Jha, 2015) and its predecessor, the Grid Application Toolkit (GAT) (Van Nieuwpoort, Kielmann & Bal, 2007). These APIs are client-side solutions rather than a server-side standardization attempt such as the Distributed Resource Management Application API (DRMAA) (Troger et al., 2007). Specifically, Xenon supports several batch schedulers such as Grid Engine, Slurm and Torque, local or remote execution on a server via SSH, as well as local or remote file access via the (S)FTP and WebDAV transfer protocols, Amazon S3 cloud storage or Apache Hadoop Distributed File System (HDFS). As shown in Figure 1B, the Xenon library translates the xenon-cli commands to scheduler-specific commands used by a target cluster. When the cluster is changed or multiple clusters are used, Xenon will simply use a different translation, while the xenon-cli commands will remain essentially the same. A user needs to modify only a scheduler type or a queue name. For this reason, xenon-cli was configured as the cluster submission command for our Snakemake-based workflow, which improved the portability by making the workflow independent of the system-specific solutions (see snakemake --cluster ‘xenon scheduler [type]’ argument).

Finally, the software reliability has been ensured by regular unit and continuous integration tests in the cloud (Travis CI, https://travis-ci.org/), which involve Docker images of different batch systems including real test data (NA12878). After successful tests, the workflow was deployed on different academic HPC systems namely the Grid Engine-based cluster at Utrecht Bioinformatics Center (UBC, https://ubc.uu.nl/), the Slurm-based Distributed ASCI Supercomputer 5 (Bal et al., 2016; https://www.cs.vu.nl/das5/) and the Dutch national supercomputer (Cartesius, https://userinfo.surfsara.nl/systems/cartesius/).

Results

The sv-callers workflow readily automated parallel execution of the tools across compute nodes and enabled streamlined data analyses for example, in a Jupyter Notebook. We tested the workflow with each WGS dataset (i.e., benchmark and cell lines) on both Grid Engine- and Slurm-based HPC systems, processing in total about 4 TiB of data. Overall, the datasets were processed at different rates on each system due to differences in data size and coverage, and system load or availability at the time of use (data not shown). Submitted jobs of the benchmark and of the cell lines datasets completed successfully (100%) in about twelve hours and in 4 days, respectively. Table 2 shows the compute resources used by the SV callers (on the UBC cluster). We found that the detection of somatic SVs required significantly more compute time or resources than that of germline SVs for the datasets used (335 vs. 41 CPU core hours). In particular, the analyses by DELLY or GRIDSS were computationally more costly per sample than those by Manta or LUMPY (see Data S1).

Table 2 Compute resources used by the SV callers.

Software	Dataset	Threads per job	Wall-clock/CPU times* (HH:MM)	Peak mem. (GiB)	Output	
Manta	Cell lines	24	01:45/14:31	5.3	94 MiB	
	Benchmark	24	00:49/05:42	2.8	225 MiB	
DELLY	Cell lines	2	19:23/25:47
66:03/85:20	3.6	13 MiB	
	Benchmark	1	00:54/00:49
09:07/08:35	0.9	28 MiB	
LUMPY	Cell lines	1	13:40/20:55	24.4	5.9 GiB	
	Benchmark	1	00:29/00:45	5.7	<1 MiB	
GRIDSS	Cell lines	24	29:31/214:13	44.4	73 GiB	
	Benchmark	24	08:24/26:04	38.5	342 MiB	
Notes:

* Timings per sample or per job are averaged using median over ten WGS (paired) samples.

DELLY calls per sample were summed over jobs of distinct SV types.

LUMPY is mostly single-threaded except its I/O related code, which results in CPU time > wall-clock time.

Further, the resulting VCF files with SV calls were checked for completeness and for overlap among the callers and sample copies. Most SVs were caller-specific while the second largest category of SVs were common to three of the four callers (Fig. 2). Importantly, merging SV calls supported by the majority of tools resulted in improved precision (but not recall) compared to the best caller according to the PacBio/Moleculo (94.8% vs. 81.6% [LUMPY]) and the Personalis/1kGP (88.1% vs. 76.8% [GRIDSS]) truth sets (Fig. 3; Data S2).

Figure 2 Venn diagrams of intersecting SV call sets.

(A) Germline and (B) somatic SVs detected in the benchmark and in the cell lines samples, respectively, using Manta, DELLY, LUMPY and GRIDSS. Most SVs are caller-specific, followed by SVs common to three of the four callers. SVs detected by the callers were filtered and merged into one set (see Section Methods). Note: The Venn diagrams include the largest GRIDSS sets as the GRIDSS output varies slightly each run using the same input. Figs. S1 and S2 show the comparisons across sample copies.

Figure 3 Precision and recall of SV detection based on a single vs. multiple caller(s) according to two truth sets.

(A) For Personalis/1kGP data, the merged calls of at least three SV callers (denoted as “merge3+”) have best precision (88.1%) while DELLY calls have best recall (85.3%). (B) For PacBio/Moleculo data, the “merge3+” calls have best precision (94.8%) while Manta calls have best recall (63%).

Conclusions

Relying on a single SV detection algorithm or caller is insufficient for comprehensive and accurate detection of SVs in cancer genomes (English et al., 2015; Fang et al., 2018; Kosugi et al., 2019). Therefore, there is an increasing need for flexible and portable bioinformatics workflows that—with minimal effort—integrate multiple SV detection and post-processing tools while making efficient use of different HPC cluster or cloud infrastructures. To address these needs, we developed the sv-callers workflow according to best practices in research software (Jiménez et al., 2017; Maassen et al., 2017; https://guide.esciencecenter.nl/). Further improvements to the workflow itself and/or its ‘backend’ are possible. For example, adding multiple read mapping tools to the workflow would enable analyses directly from raw sequencing data. Moreover, the SV reporting step could be enhanced with interactive visualization to facilitate manual inspection of SVs in genomic regions of interest. Furthermore, the SV callers’ binaries (currently distributed via bioconda) are suboptimal regarding the performance for target systems used, and therefore the codes might benefit from further optimization (e.g., using the Intel C/C++ Compiler). We aim to extend cloud support in Xenon with schedulers such as Microsoft Azure Batch and Amazon AWS Batch as well as to improve the coupling between Xenon and Snakemake software. Finally, the sv-callers workflow is an open-source software that is freely available from the ELIXIR’s bio.tools registry (https://bio.tools/sv-callers) and Research Software Directory (https://research-software.nl/software/sv-callers).

Supplemental Information

Supplemental Information 1 UpSet plots of intersecting SV call sets detected by each caller using the benchmark sample copies.

SV call sets detected by Manta (A), DELLY (B) and LUMPY (C) overlap 100% except those by GRIDSS (D).

Click here for additional data file.

Supplemental Information 2 UpSet plots of intersecting SV call sets detected by each caller using the cell lines sample copies.

SV call sets detected by Manta (A), DELLY (B) and LUMPY (C) overlap 100% except those by GRIDSS (D).

Click here for additional data file.

Supplemental Information 3 Wall-clock/CPU times of germline and somatic SV calling on the benchmark and cell lines datasets, respectively.

Click here for additional data file.

Supplemental Information 4 Evaluation of SV calls according to two truth sets.

Manta, DELLY, LUMPY, GRIDSS calls vs merged calls that agree by at least two or three callers (denoted as ’merge2+’ or ’merge3+’, respectively).

Click here for additional data file.

The authors would like to thank HPC system administrators and SURFsara consultants in particular Jeroen Engelberts and Lyke Voort for technical support, and the reviewers for their valuable suggestions.

Additional Information and Declarations

Competing Interests

Author Contributions

Data Availability

Jeroen de Ridder is co-founder of Cyclomics B.V.

Arnold Kuzniar conceived and designed the experiments, performed the experiments, developed the software, analyzed the data, prepared figures and/or tables, authored or reviewed drafts of the paper, and approved the final draft.

Jason Maassen (co)developed the software, analyzed the data, prepared figures and/or tables, authored or reviewed drafts of the paper, and approved the final draft.

Stefan Verhoeven (co)developed the software, reviewed drafts of the paper, and approved the final draft.

Luca Santuari contributed to the software, analyzed the data, performed the experiments, prepared figures and/or tables, authored or reviewed drafts of the paper, and approved the final draft.

Carl Shneider contributed to the software prototype, reviewed drafts of the paper, and approved the final draft.

Wigard P. Kloosterman conceived and designed the experiments, reviewed drafts of the paper, and approved the final draft.

Jeroen de Ridder conceived and designed the experiments, authored or reviewed drafts of the paper, and approved the final draft.

The following information was supplied regarding data availability:

The source code including test data were deposited at the Zenodo archive (DOI 10.5281/zenodo.1217111 and DOI 10.5281/zenodo.2663307).

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
