# Peer review of "sv-callers: a highly portable parallel workflow for structural variant detection in whole-genome sequence data"

_PeerJ, doi:10.7717/peerj.8214_

## Round 0.1 · original submission · Major Revisions

Dear Dr. Kuzniar and colleagues:

Thanks for submitting your manuscript to PeerJ. I have now received two independent reviews of your work, and as you will see, the reviewers raised some concerns about the research. Despite this, these reviewers are optimistic about your work and the potential impact it will have on research communities studying informatics analysis of short-read sequencing data. Thus, I encourage you to revise your manuscript, accordingly, taking into account all of the concerns raised by both reviewers.

The reviewers struggled with the content and scope of your manuscript. Please ensure that your program is applicable to the wider audience as suggested by your title. Please also ensure that all of the content is accessible to the reviewers. Use the suggestions from both reviewers to improve the organization and clarity of your findings.

While the concerns of the reviewers are relatively minor, this is a major revision to ensure that the original reviewers have a chance to evaluate your responses to their concerns.

I look forward to seeing your revision, and thanks again for submitting your work to PeerJ.

Good luck with your revision,

-joe

Reviewer 1 ·

Basic reporting

No comment

Experimental design

No comment

Validity of the findings

No comment

Additional comments

The authors propose the workflow for structural variant detection in whole-genome sequence data. It is necessary and useful, however, there are some concerns that need to be addressed:
- The system can detect general variants or specific variants for cancer? Because as shown in the title, it maybe detect the general variants, but in the text, the authors mentioned a lot in cancer.
- Can the system classify the structural variants (i.e., deletion, insertion, ...)?
- The system can be extended to be used in the other operating system apart from Linux. It is necessary to have broader users.
- As many similar systems can support the use of GPU in running, can we config the "sv-caller" for using GPU?
- The most important thing is that the authors have to provide more comparisons and discussions on the previous workflows. There are a lot of published systems for variant calling, what are the advantages and disadvantages of this workflow?
- I have checked that I cannot access the link of Research Software Directory (https://research-software.nl/software/sv-callers), please check that it is always active.

Reviewer 2 ·

Basic reporting

The authors has implemented an SV discovery workflow that integrated multiple publicly accessible SV algorithms for easy application. This tool is reliable and extendable, making it easier for users to generate SV calls. However, assessment of the tool, as represented in the manuscript, were mostly focused on the computational efficiency but lack discussion about quality of the SV callset generated from this tool. The authors should have provided a comprehensive evaluation of the tool, with comparison against individual algorithms, based on the two samples discussed in the manuscript to show superiority. Moreover, the content is not well organized in this manuscript as is confusing to read. Serious re-format is necessary here.

Experimental design

Whole design of this manuscript focused on improving computational efficiency of SV calling algorithms, but didn't provide enough details about important aspects such as 1. SV comparison criteria adopted 2. benchmark the proposed method to other existing ones

Validity of the findings

Kuzniar et al aimed at building a reliable tool for SV discovery in germline and cancer genomes. They proved reliability of their tool, however, failed to provide enough evidence that the performance of their workflow are of high sensitivity and low false discovery rate.

Additional comments

Major point:

Kuzniar et al. mostly focused on building a user-friendly, high efficient workflow for SV discovery via integration of different SV calling methods, however, little results were provided about the quality of output SV callset. These following points should have been addressed in both method and result :
1. explain the criteria adopted for the comparison of SVs from different algorithms;
2. provide evaluation of SVs from each algorithms, and explain their strategy of filtering SVs.
3. Most importantly, the authors should provide enough evaluation of the callset from their workflow, show the increase in sensitivity and specificity of their workflow compared against others.

The method part includes information that should have been moved out to other sections. Eg. The ‘Workflow implementation and deployment’ section under method need serious re-writing, the introduction and discussion of snakemake would fit better under ‘Introduction’, while the discussions about figure 1 should be under ‘methods’.


Minor points:
1. In line 176, author mentioned that time cost of the tool on different system differ substantially, and this difference were contributed by data size and load of the computing system. This reads confusing, as for a fair comparison the same dataset should have processed then why is there a difference in data size? And how many jobs were processed in parallel on each system?
2. Which computing system were number in table 2 concluded from?

---

## Round 0.2 · Minor Revisions

Dear Dr. Kuzniar and colleagues:

Thanks for resubmitting your revision. Reviewer 2 has raised a few concerns and I would like you to address these.

I look forward to seeing your revision, and thanks again for submitting your work to PeerJ.

Good luck with your revision,

-joe

Reviewer 1 ·

Basic reporting

No comment

Experimental design

No comment

Validity of the findings

No comment

Additional comments

No comment

·

Basic reporting

The authors present sv-callers, a computational workflow managing execution of multiple SV-detection tools for ensemble analysis of a given NGS dataset. The code is made publicly available and free to use without limitations. This manuscript includes links to example output from the software, along with a Jupyter Notebook for others to reproduce their results. The software is readily configurable for additional SV-callers and runs to completion on multiple HPC infrastructures. The manuscript's narrative is clear and easy to read. An attentive reader of this manuscript will come away with awareness of a new tool available for use in their research.

Experimental design

There is a need in the community for workflows like sv-callers. The authors cite competing tools by English, et al. (2015) and Fang, et al. (2018), The authors claim that sv-callers "enables analyses that would otherwise be difficult and/or slow to perform." While this may be true, the manuscript provides no evidence supporting this conclusion. Otherwise, the questions answered by the authors address runtime, resource utilization, and whether the output is usable in downstream applications (i.e. R), none of which qualify as meaningful and relevant research questions in the fields of Biological, Environmental, Medical, or Health Sciences Sciences.

Validity of the findings

no comment

Additional comments

Reviewer 2 gave the authors an opportunity to augment their findings with an interpretation of their results in light of a research question, i.e. "how accurate are the merged calls?" In response, the authors demurred, suggesting an effort of massive proportions (e.g. Kosugi et al.) would be required. Even if the authors manually review 20 deletions contained in a NA12878 SV truth set (e.g. Parikh H et al. BMC Genomics. 2016 Jan 16;17:64) some semblance of an answer would emerge with allowances for uncertainty based on the small sample size.

This reviewer hopes the authors will make the suggested (or equivalent) effort, and increase visibility of their software via publication in PeerJ.

---

## Round 0.3 · accepted · Accept

Dear Dr. Kuzniar and colleagues:

Thanks for re-submitting your revised manuscript to PeerJ, and for addressing the concerns raised by the reviewers. I now believe that your manuscript is suitable for publication. Congratulations! I look forward to seeing this work in print, and I anticipate it being an important resource for research communities studying informatics analysis of short-read sequencing data.

Thanks again for choosing PeerJ to publish such important work.

-joe